# Ageing and Olfactory Dysfunction in Trisomy 21: A Systematic Review

**DOI:** 10.3390/brainsci11070952

**Published:** 2021-07-20

**Authors:** Hanani Abdul Manan, Noorazrul Yahya

**Affiliations:** 1Makmal Pemprosesan Imej Kefungsian, Department of Radiology, Faculty of Medicine, Pusat Perubatan Universiti Kebangsaan Malaysia, Jalan Yaacob Latif, Bandar Tun Razak, Cheras, Kuala Lumpur 56000, Malaysia; 2Diagnostic Imaging & Radiotherapy Program, Faculty of Health Sciences, Universiti Kebangsaan Malaysia, Jalan Raja Muda Abdul Aziz, Malaysia, Kuala Lumpur 50300, Malaysia; azrulyahya@ukm.edu.my

**Keywords:** trisomy 21, Down syndrome, olfactory dysfunction

## Abstract

Purpose: The olfactory system is particularly vulnerable in an ageing brain, both anatomically and functionally, and these brain changes are more pronounced among individuals with trisomy 21. Furthermore, the age of the system starts to deteriorate, and the mechanism involved is unclear in an individual with trisomy 21. Therefore, the present review aims to summarise the available information related to this topic and to suggest questions still unanswered which can be a subject of further research. Methods: A systematic literature search of trisomy 21 and olfactory dysfunction was conducted using PubMed/MEDLINE and Scopus electronic database following PRISMA guidelines. References and citations were checked in the Google Scholar database. Reports were extracted for information on demographics and psychophysical evaluation. Then, the reports were systematically reviewed based on the effects of ageing on the three olfactory domains: threshold, discrimination, and identification. Results: Participants with trisomy 21 show an early onset of olfactory impairment, and the age effect of the olfactory deficit is fully expressed at age > 30 years old. The three olfactory domains, threshold, discrimination, and identification, are suggested to be impaired in trisomy 21 participants with age > 30 years old. Conclusions: Olfactory dysfunction in an individual with trisomy 21 commences at a relatively young age and affects the three olfactory domains. A challenge for the future is to quantitatively establish the olfactory function of an individual with trisomy 21 at all ages with more detailed measurements to further understand the pathophysiology of this brain deterioration.

## 1. Introduction

Trisomy 21, or Down syndrome (DS), is a genetic disorder affected by the manifestation of all or part of the third copy of chromosome 21 (trisomy 21; +21) and is the most common congenital chromosome disorder in humans [1]. Each year, about 6000 babies are born with trisomy 21; about 1 in every 700 babies are born [2]. The incidence of trisomy 21 among the Malaysian population is about 1 in every 959 babies born [3], lower than those from Western populations. Trisomy 21 is unique among human diseases producing a viable, functional human being with an autosomal chromosome’s triplication. Individuals with trisomy 21 manifest intellectual and physical delay, and dysmorphic facial features and/or congenital heart malformations [4,5], susceptibility to leukaemia [6], and infections. The most consistent differences, however, involve the brain [7]. These differences are still incompletely understood, causing a developmental abnormality that results in lifelong cognition differences [7]. On top of that, individuals with trisomy 21 are also associated with a group of clinical manifestations of ‘accelerated ageing’ [8,9] in which individuals with trisomy 21 will age faster than the general population. It is expected that adults with trisomy 21 will show physical, medical, and cognitive signs of ageing much earlier than what is expected for their age [2].

The life expectancy of an individual with trisomy 21 increased dramatically between 1960 and now. In 1960, an individual with trisomy 21 had a mean survival of about 10 years old, which increased to 47 years in 2007 [10]. Early in this century, the mean survival was 61.1 years for males and 57.8 years for females in Australia [4]. Therefore, the effects of this accelerated ageing in individuals with trisomy need to give more attention due to increased life expectancy. A review by Zigman in 2013, suggested that accelerated ageing in trisomy 21 is atypical and segmental [9]; this involves only some of the organs and tissues, including the brain [9]. The present review is interested in summarising the effects of this accelerated ageing on the olfactory system and addressing at what age the olfactory system starts to deteriorate in trisomy 21. Furthermore, the present review also aims to address the hypothesis that adults with trisomy 21 show greater deficits in olfactory function than younger adults with trisomy 21. Moreover, in an extensive review on trisomy 21 and ageing [9], the olfactory dysfunction was not considered, suggesting that research is needed to explore the olfactory function pattern across the whole life span in individuals with trisomy 21. Previous studies investigating olfactory function in trisomy 21 is limited [11,12,13,14,15,16,17,18,19,20] and lacking a comprehensive evaluation. Therefore, an investigation into the nature of olfactory deficit in individuals with trisomy 21 is of interest and, therefore, suggests unanswered questions that can be a subject of further research.

Previous studies have shown that adults with trisomy 21 between the ages of 20 and 40 years old develop symptoms of pathologic changes in the brain and neuropathology similar to that of Alzheimer’s disease (AD) [10,21,22]. The same characteristic of senile plaques in AD has been found in trisomy 21 as early as 1929 [23]. Clinical deterioration similar to AD in trisomy 21 has been known since 1948 [23]. In the 1960s, the two disorders had been linked [21]. Olfactory dysfunction is an early symptom of dementia, including AD [24]. It has a relatively high prevalence in various types of dementia, reaching up to 100% in AD, 90% in Parkinson’s disease dementia, 96% in frontotemporal dementia (FTLD), and 15% in vascular dementia [25,26]. Previous studies proposed that olfactory dysfunction in AD originates from olfactory epithelium (OE) [27,28]. This is based on animal studies, which found cells that comprise the OE and the pathways for transmitting olfactory sensations to the olfactory bulb and other sites are involved in processing these chemical sensations [24,29,30]. In mammals, olfactory neurons die and are continually replaced due to olfactory mucosa cells’ ability to generate new populations of sensory neurons throughout the lifespan [31,32]. However, AD shows functional changes that accompany dysfunction in olfactory areas, both peripherally and centrally. These changes occurring in the OE lead to neurodegeneration, which seems to be cell-autonomous and independent of plaque accumulation [33]. Odorant responses in the OE are also reduced in several mouse models of AD, similar to the decreased olfactory ability observed in humans [34].

Previous works showed olfactory impairment in individuals with trisomy 21; however, the age of these olfactory dysfunctions and the start of brain pathology, and the mechanism involved is unclear. These changes in the early stages of olfaction have the potential to be an inexpensive and non-invasive diagnostic tool and biomarker in individuals with trisomy 21. Therefore, the present review aims to summarise at what age this olfactory dysfunction starts to develop in trisomy 21 and gather available information related to this topic, and may suggest questions still unanswered, which can be a subject of further research.

## 2. Methods

### 2.1. Search Strategy and Study Selection

A systematic search was conducted by two independent researchers (HAM and NY) in PubMed and Scopus electronic databases. The systematic search method used in the present study was based on the Preferred Reporting Items for Systematic Reviews and Meta-Analyses guidelines (PRISMA) [35,36] and followed previous studies [37,38,39,40,41,42]. The search was performed to identify studies reporting trisomy 21 or Down syndrome and olfactory dysfunction using a psychophysical and physiological measure. Article search was conducted between the earliest record and 17 June 2021. Search terms were as follows: trisomy-21, trisomy 21, down syndrome, down-syndrome, olfaction, olfactory, smell system, and smell. We also manually checked for related articles in references and citations through the Google Scholar database. There was no limitation on publication status or publication date. All records were grouped into a final database after removing duplicates, followed by screening by titles and abstracts by HAM and NY, independently. Consensus for eligibility was reached through discussion; the information was tabulated in Figure 1. We used an assessment tool from the National Heart, Lung and Blood Institute, Quality Assessment Tool for Observational Cohort and Cross-Sectional Studies, to assess the quality of included studies (https://www.nhlbi.nih.gov/health-topics/study-quality-assessment-tools, accessed on 23 June 2021).

### 2.2. Inclusion Criteria and Exclusion Criteria

Original studies in English reported in peer-reviewed journals describing trisomy 21 or Down syndrome and olfactory dysfunction using psychophysical and physiological measurements were selected. The studies on humans describing and assessing any components of the olfactory such as olfactory bulb (OB) volume, olfactory sulcus (OS) depth, and Threshold–Discrimination–Identification (TDI) score were included. We excluded all review articles as well as case reports and case series studies. Following the removal of duplicates, those evidently outside the scope of the review were rejected.

From the eligible studies, the following variables were recorded: year of publication, country, study author(s), analysis mode, participants’ demographics, including age, handedness, duration of olfactory loss, psychophysical and physiological tests, and principal findings.

## 3. Results

### 3.1. Study Demographics and Details

Table 1 summarises the demographic information of individuals with trisomy 21 and the olfactory system of seven studies. Generally, studies have reasonable quality, as shown in Appendix A. Sample size calculations were varied between articles; however, all the studies reported more than 20 participants. A sample size of 20 participants are able to detect a significant olfactory impairment in an individual with trisomy 21, and power analysis indicated a power of 85% to detect a significant difference in impairment [12,43]. None of the studies was blinded due to the nature of the studies, which require the direct involvement of personnel-in-charge. All studies reported cross-sectional case-control study, and none of the studies reported longitudinal study. Across seven studies, 544 participants were investigated; details are tabulated in Figure 1. A total of 234 participants had trisomy 21. Studies also included 310 control group participants, and this number including 90 participants with the same mental capacity with trisomy 21 participants. Gender of participants were reported in some of the studies and indicated slightly more males (trisomy 21: males = 84, females = 65 and not mentioned = 85, HC: male = 92, female = 89 and not mentioned = 129). The age of the participants ranged from 9 to 57 years old. All of the studies compared participants with trisomy 21 to age-and sex-matched HC, and three studies also compared participants with the same mental capacity with trisomy 21 participants [13,14,18]. Two studies conducted separate analyses of age’s effects on olfactory function [13,18]. The studies found that olfactory dysfunction is more pronounced in older participants, and their odour sensitivity deficits are apparent within an individual with trisomy 21 with advancing age [18]. One study conducted separate analyses based on gender and found that females with trisomy 21 have a lower number of olfactory dysfunction than males over a large age range [11].

All studies used validated measures to measure olfactory outcomes. One study used a standardised measure of olfactory function in their cohort, the ‘Sniffin Sticks’. The test is composed of three parts: odour threshold for n-butanol or phenyl ethyl alcohol (T), odour discrimination (D), and odour identification (I). The final score was expressed as the sum of the T, D, and I value. Three studies used the University of Pennsylvania Smell Identification Test (UPSIT) [44]. UPSIT [44] is a standardised multiple-choice identification test that incorporates 40 microencapsulated odourants into an easily administered format and is the most common means for assessing olfactory function in North America [45,46]. One study used The San Diego Odor Identification Test [17], and one study used a combination of UPSIT and The San Diego Odor Identification Test. All studies combined the standardised measure of olfactory function with other questionnaires or assessments such as the Nasal Health Questionnaire, matching and naming olfactory task, Wechsler Adult Intelligence Scale—Revised (WAIS—R) [47], The Wechsler Intelligence Scale for Children—Revised (WISC—R) [48], the picture identification test (PIT) and the Peabody picture vocabulary test—revised (PPVT—R) [49] and The Dementia Rating Scale [43]. Three studies also evaluated the intelligence using a test such as Stanford–Binet Intelligence Scales [50]. One study assessed the anterior rhinoscopic upper airway by an experienced otorhinolaryngologist to ensure nasal cavities were patent and to rule out diseases that might preclude proper olfactory testing. Details of the psychophysical, physiological tests, and questionnaire used are tabulated in Table 1.

### 3.2. Threshold–Discrimination–Identification Scores

Even though all the studies reported that they used validated measures to measure olfactory outcomes, only one study reported the Threshold–Discrimination–Identification (TDI) score, and one study reported the odour threshold score. Even though the reported scores are from different olfactory function tests, the result shows a significant olfactory function deficit compared to HC. Table 2 also tabulates the list of nasal chemosensory tests used for each study. The present review found that all studies used a different set of tasks and combinations to evaluate the olfactory function. In the present study, we note that the measurement used to measure psychological scores were different. However, we expect that the differences between the tests used are small and within an acceptable range.

### 3.3. Olfactory Dysfunction in Trisomy 21

A detailed summary of the main findings was tabulated in Table 3. Results demonstrate that: (1) All studies are in agreement that participants with trisomy 21 show early onset of olfactory dysfunction, which is fully expressed at age 30. The olfactory impairment is more significant in older individuals, and a previous study suggested a progressive impairment over time, suggesting the age effect [18]. However, one study [11] indicates that younger adult individuals with trisomy 21 are severely impaired. The most probable reason is due to Cecchini et al., who reported trisomy 21 with age > 18 years old. On the contrary, one study evaluates participants with trisomy 21 with mean age: 13.89 ± 1.98 years old and found the olfactory function is comparable with HC. Furthermore, the study suggested that young adolescents with trisomy 21 do not exhibit decreased olfactory function relative to HC matches based on intellectual function [14]. On top of that [18] also shows a similar finding with McKeown et al., and Nijjar and Murphy, also reported individuals with trisomy with mean age 14.5 ± 1.79 years old. (2) Olfactory function (olfactory threshold, olfactory discrimination, and olfactory identification) is suggested to be impaired in participants with trisomy 21 with age > 30 years old, and this impairment increased with increasing age. Most of the selected studies used the odour identification test. All the studies that used the test reported impairment in odour identification. This is followed by the odour threshold test, which also shows impairment in an individual with trisomy 21. Finally, only two studies used the odour discrimination test, and both studies found impairment in odour discrimination [11,14]. The reason why the studies do not use all three domains of the test was not mentioned. Two studies also reported participants with trisomy 21 were impaired in odour recognition [13,37]. (3) One study also evaluated the taste threshold and observed that participants with trisomy 21 show comparable performance in a taste threshold task similar to the olfactory threshold task in HC [16]. Exclusion criteria also indicate to exclude trisomy 21 individuals who are mentally impaired in understanding the directions involved in one or more of the tasks [16,18]. Therefore, these suggest that the trisomy 21 participants’ poor performance was not due to task demands. (4) One study reported that females with trisomy 21 have less impaired olfactory function than males over a broad age range [11]. Finally, (5) one study reported that trisomy 21 participants did not seem aware of their olfactory status, and they self-reported to have a normal sense of smell [11].

### 3.4. Olfactory Dysfunction in Trisomy 21 and the Relation to Early Onset of Alzheimer?

The present review observes that odour identification is the most reported, and all studies reported impaired odour identification in trisomy 21 participants. However, it is unclear why most of the studies used odour identification in evaluating olfactory function in trisomy 21 individuals and not other domains. The present review hypothesises that this might be due to the odour identification being the easiest to administer by the experimenters. Or this could also be because this domain is related to olfactory dysfunction in AD, and the first domain shows the deterioration. This specific olfactory identification impairment is similar to that seen in AD patients [13]. A study suggested that the impaired ability of trisomy 21 participants to recognise and match the target odours to the previously presented odours, as compared to the HC, was likely due to the deterioration in the olfactory system as well as the structures involved in facilitating memory [16]. The study further suggests that there may be other functional manifestations of the neuropathology developing in the brain of the older person (age > 30 years old) with trisomy 21 in addition to dementia. Additionally, [14] reported that olfactory dysfunction in an individual with trisomy 21 occurs only at ages when AD-related pathology is just beginning to develop. A detailed summary of the present findings was tabulated in Table 4.

It is important to note that only one study reported a dementia score [16]; therefore, the correlation between olfactory dysfunction and dementia score cannot be ascertained. The present review would also like to highlight that the reviewed studies mentioned senile plaques and neurofibrillary tangles changes, but none of the studies evaluates these parameters among participants with trisomy 21 and only speculated on the association based on findings from post-mortem examinations of individuals with trisomy 21 studied.

## 4. Discussion

This present review is the first systematic review summarising the olfactory function in individuals with trisomy 21. The most important finding is that individuals with trisomy 21 demonstrate an early onset of olfactory dysfunction, >30 years old, involving all three domains; odour threshold, odour discrimination, and odour identification. The present review also would like to highlight the previous studies’ observation, suggesting that individuals with trisomy 21 demonstrate similar neuropathology to Alzheimer’s disease (AD) and olfactory dysfunction could be the early indication of AD in trisomy 21.

### 4.1. Olfactory Dysfunction and Trisomy 21

Most of the studies agree that olfactory dysfunction in trisomy 21 starts early but with a threshold. There is an age effect on this olfactory dysfunction. For example, a study by [14] reported participants with a mean age of 13.89 ± 1.98 years old and found the olfactory system is comparable with healthy control (HC). However, the recently published study suggests that younger individuals with trisomy 21 are severely impaired. Cecchini et. al. reported participants with age > 18 years old, and this study does not include the cognitive ability test of the participants [11]. In the present review, we proposed the differences in the finding between Cecchini et al., and most of the study might be due to the age factor, as participants in [14,18] are very young with a mean age of around 14.5 years old. The other probable explanation could be due to participants’ inability to understand the directions involved in one or more tasks. This could be a cause of increased impairment in young individuals with trisomy 21. With this available evidence from the previous studies, the present review would like to suggest that the olfactory dysfunction in an individual with trisomy 21 is present early and begins to accumulate before individuals are 30 years old. We would also like to suggest further that the olfactory dysfunction in an individual with trisomy 21 is fully expressed at age > 30 years old, and results demonstrate a significant reduction in odour threshold, odour discrimination, and odour identification.

In the present review, we found that most of the studies did not evaluate the combination of all three domains (odour threshold, odour discrimination, odour identification), but instead only evaluate one or two domains. The reason they choose either one or a combination of two domains from the three domains is unclear. Most of the studies used odours identification; perhaps odours identification appears to be the first domain altered in trisomy 21 [14,52] or due to strong linkage of odour identification with AD [53] (discussed in details below). This is followed by the odour threshold, and only two studies reported odour discrimination, and both studies demonstrate significant deficit [11,14]. This result demonstrates that an individual with trisomy 21 with age > 30 years old has an impaired ability to detect odours. Important to note, participants with trisomy 21 self-reported to have a normal sense of smell, even though the psychophysical results show all three domains, olfactory odour threshold, odour discrimination, and odour identification, are impaired. This suggests that participants are not aware of their olfactory status. One possible explanation for this situation might be due to a slowly progressive of this olfactory dysfunction. This trend of gradual smell loss was also reported in patients with AD and Parkinson’s disease (PD) [54]. Therefore, we would like to suggest that individuals with trisomy 21 and caregivers must be educated to appreciate the relevance of olfaction in daily life, especially in recognising dangerous odours, such as gas, smoke, or spoiled food [11]. We would further suggest a regular check-up to detect this olfactory dysfunction.

### 4.2. Cause of Olfactory Impairment in Individuals with Trisomy 21

The olfactory impairment in individuals with trisomy 21 is suggested to be due to peripheral changes at the mucosal level, in the olfactory bulb, or in the olfactory tract [24,55]. Fitzgerald et al., 2013, reported that an individual with trisomy 21 had shown a high incidence of upper-respiratory infections and nasal itching; this is proposed due to the airways’ morphologic variations [55]. Meanwhile, Zou et al., 2016 reported that the incident of hospitalisation of an individual with trisomy 21 due to upper-respiratory infections is also high [24]. Contrary to the previous reports, the results of this review reported different findings, the differences of an incident of upper-respiratory infections were not significant, and nasal health is comparable in an individual with trisomy 21 and HC. Based on the report of the selected studies, the present review suggests that nasal dysfunction is unlikely to contribute to olfactory impairment in an individual with trisomy 21 [12]. However, this aetiology needs to be investigated further with a more sophisticated neuroimaging technique. An increased incidence of and long-term effects of nasal sinus disease and allergic rhinitis should also be ruled out.

### 4.3. Odour Identification, Trisomy 21, and Alzheimer’s Disease

Most of the studies reported odour identification and observed olfactory identification impairment in an individual with trisomy 21. The present review proposes that the odour identification domain is not the best way to investigate olfactory function in people with trisomy 21. Odour identification is the easiest to administer for the experimenter. The test is strongly verbally confounded, which is a big minus for the studied population. Therefore, for future study, we would like to propose for interviews, the evaluation of odour thresholds, and using olfactory event-related potentials for imaging.

On the other hand, studies of patients with AD also found a similar observation: reduction in odour identification [56,57,58]. It was reported that odour identification appears to be the most altered in AD [59]. It is essential to point out that, even in healthy elderly groups, there is a reduction in olfactory sensitivity. However, the different domains were affected; odour discrimination (olfactory thresholds) is the most affected in healthy elderly groups [60]. We need to consider that olfaction is also correlated with recall mechanisms due to its synchronisation with the hippocampus in creating and retrieving olfactory associative memory [61]. These differences between people with AD and healthy elderly can be explained by the association of execution and cognitive memory domains, related in part to performance on tests that involve identification and recognition, closely related to semantic memory [62]. The previous study proposed that the possibility of neuropathological changes is occurring in the older individual (>30 years old) with trisomy 21 in the same regions in which they appear in patients with AD [20]. Those areas which mediate olfactory functioning, such as the entorhinal cortex, piriform cortex, and the anterior olfactory nucleus, may be particularly vulnerable to ageing [13,14,20]. However, based on the similarity in olfactory identification impairment, there is insufficient evidence to correlate these neuropathological changes among two groups of participants: individuals with trisomy 21 and AD patients. Most of the selected studies are lacking in structural and functional details. Therefore, the present review would not be able to conclude that the neuropathology seen in an older individual with trisomy 21 has similar characteristics with people with AD. Further research, including structural and functional imaging, is necessary to determine the severity of the impairment. Measures of current cognitive functioning are also needed to detect the early manifestation of dementia in older individuals with trisomy 21. A longitudinal design would provide unambiguous evidence of increasing olfactory identification deficits with advancing age and dementia in individuals with trisomy 21.

### 4.4. Limitations

The present review highlights that most of the studies were old, and some studies are more than 20 years old. There are limited numbers of trisomy 21 and olfaction studies in the last 5 years, and assessment methods were not contemporary. All studies lack details and only reported nasal chemosensory results. None of the studies reported structural (such as olfactory bulb, olfactory sulcus, grey matter, or white matter) and functional changes through brain imaging. Until recently, our understanding of the structural brain abnormalities in individuals with trisomy 21 was almost exclusively based on autopsy studies; improvements in magnetic resonance imaging (MRI) and image-processing techniques allowed quantitative explorations of brain structure in living subjects with trisomy 21. Even though there are multiple limitations in the articles reviewed, we hope that this review will spark a new interest for more trisomy 21 and olfaction research in the future.

## 5. Conclusions

The individual with trisomy 21 shows the early onset of olfactory dysfunction, >30 years old. This olfactory dysfunction is involved in three domains: odour threshold, discrimination, and identification. A challenge for the future is to quantitatively establish the olfactory function of an individual with trisomy 21 at all ages with more detailed measurements, for example, olfactory bulb, olfactory sulcus, grey matter, and white matter. Moreover, further work involving, e.g., structural and functional magnetic resonance imaging is required to correlate olfactory dysfunction and brain pathology in trisomy 21 more precisely. This is to establish and determine the pathological basis for the losses.

## Figures and Tables

**Figure 1 brainsci-11-00952-f001:**
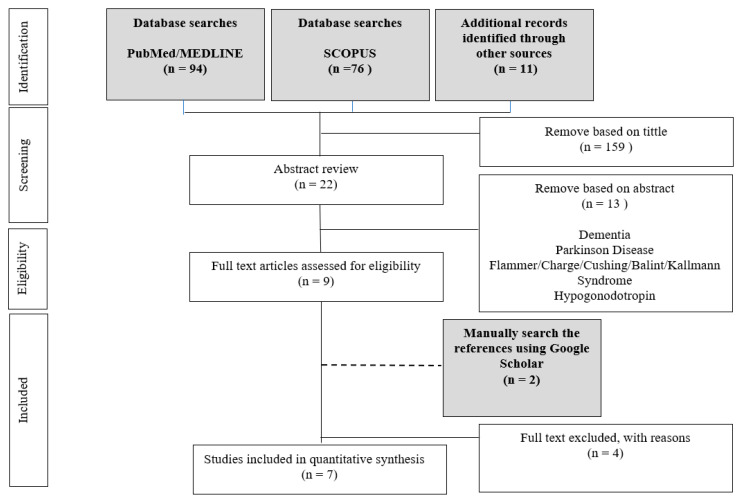
Diagram of the search process for studies included in the present systematic review.

**Table 1 brainsci-11-00952-t001:** Studies of the olfactory system and individual with trisomy 21: demographic information, clinical characteristics, and associated tests.

Author(s)	Age and Gender	Psychophysical, Physiological Tests and Questionnaire	IQ Score	Notes	Research Design
Cecchini et al., 2016 [11]	56 T-21 (31M/25F) 18–57 years 56 HC (31M/25F)	Sniffin’ Sticks WAIS—R. The verbal tests are Information, Comprehension, Arithmetic, Digit Span, Similarities, and Vocabulary. The Performance subtests are Picture Arrangement, Picture Completion, Block Design, Object Assembly, and Digit Symbol. VIQ, PIQ and TIQ * only 13 T-21 individuals have a cognitive evaluation		All T-21 participants reported having a normal sense of smell	Cross-sectional case–control study
Murphy and Jinich, 1996 [16]	23 T-21 25–46 years Mean age: 30.1 ± 7.2 23 HC 21–46 years Mean age: 30.0 ± 7.3	UPSIT The Dementia Rating Scale (level of cognitive functioning)		Threshold test: n-butyl alcohol Identification test: 40-item UPSIT Odour memory test: common odours are drawn from a battery of 80 (coffee, spearmint, peanut butter)	Cross-sectional case–control study
McKeown et al., 1996 [14]	20 T-21 (11M/9F) Mean age: 13.89 ± 1.98 20 HC 20 control with same mental capacity with T-21	UPSIT 16-item, three-choice test of odour discrimination which does not require verbal identification of the stimuli (Smith et al., 1993 [19]) Picture identification test and Peabody picture vocabulary test—revised (cognitive test)		Otorhinolaryngological Evaluation Anterior rhinoscopic upper airway evaluation	Cross-sectional case–control study
Chen et al., 2006 [12]	20 T-21 14–41 years Mean age: 26.00 16 HC 19–38 years Mean age: 26.00	The San Diego Odor Identification Test (Murphy et al., 2002 [17]) Nasal health questionnaire		Olfactory threshold and odour identification tests	Cross-sectional case–control study
Nijjar and Murphy, 2002 [18]	67 T-21 18 Child (17M/7F) 14.5 ± 1.79 23 YA (10M/13F) 23.2 ± 3.95 26 adults (15M/11F) 38.8 ± 7.07 70 HC 22 Child (10M/12F)14.5 ± 2.41 31 YA (15M/16F) 24.2 ± 2.84 17 adults (11M/6F) 40.0 ± 7.26 55 control with same mental capacity with T-21 18 Child (6M/12F) 15.3 ± 1.14 15 YA (8M/7F) 21.6 ± 3.60 22 adults (11M/11F) 42.5 ± 8.19	The San Diego Odor Identification Test UPSIT Task orientation: *to determine whether the subject had a good comprehension of the experimental task*	67 T-21 18 Child, 47.9 ± 4.26 23 YA, 51.5 ± 7.89 26 adults, 53.5 ± 9.37 55 control with same mental capacity with DS 18 Child, 52.9 ± 10.51 15 YA, 52.6 ± 9.38 22 adults, 56.9 ± 11.61	Threshold test: n-butyl alcohol Odour threshold administration: Picture-based odour identification test: The San Diego Odor Identification Test Picture-Based Odor Identification Administration: Lexical-based odor identification test: UPSIT	Cross-sectional case–control study
Hemdal et al., 1993 [13]	20 T-21 9–49 years Mean age: 23.1 21 HC Mean age: 23.8 15 control with same mental capacity with T-21 10–35 years Mean age: 20.6	M-UPSIT WISC or WISC—R full scale	IQ range DS: 35 and 56 means score: 45.85 IQ range control with same mental capacity with DS: 35 and 72 Mean score: 49.87		Cross-sectional case–control study
Zucco and Negrin, 1994 [20])	28 T-21 14 YA 20–31 years 26.5 ± 3.0 14 adults 32–54 years 41.0 ± 6.8	Matching and naming olfactory task	YA: Stanford–Binet Mental Age = 5.6 ± 1.1 Adults: Stanford–Binet Mental Age = 5.1 ± 0.9	Required to recognise among 4 sniffed odour, while on the latter had to label an odour by choosing among four alternatives None showed symptoms of dementia	Cross-sectional case–control study

T-21 = individual with trisomy 21, HC = healthy control, YA = young adult, OB = olfactory bulb, OS = olfactory sulcus, GM = gray matter, UPSIT = University of Pennsylvania Smell Identification Test, M-UPSIT = University of Pennsylvania Smell Identification Test, WAIS—R = Wechsler Adult Intelligence Scale—Revised, VIQ = Verbal Intelligence Quotient, PIQ = Performance Intelligence Quotient, TIQ = full-scale IQ [14], * Sinonasal endoscopy was evaluated to exclude obstructive or inflammatory causes of smell dysfunction.

**Table 2 brainsci-11-00952-t002:** Psychophysical result of the individual with trisomy 21 and healthy control group.

Author	T-21				HC				Notes	Nasal Chemosensory Test
	Psychophysical Tests				Psychophysical Tests					
	Total	T	D	I	Total	T	D	I		
Cecchini et al., 2016 [11]	16.7 ± 5.13 (range 5.0–27.5)		5.9 ± 1.97	7.8 ± 2.81	35.4 ± 3.75 (range 27.2–46.0)		13.0 ± 1.77	13.9 ± 1.45	A total of 27 T-21 individuals (20M, 7F) out of 56 showed functional anosmia (i.e., TDI < 16).	Sniffin’ Sticks Threshold: the concentration at which the odour is reliably detected Discrimination: the ability of the subject to distinguish odours Identification: the identification of different odours from a list of odours
Murphy and Jinich, 1996 [16]										Threshold: odorant n-butyl alcohol and Sucrose 0.18 M + the 40-item UPSIT Identification + Memory for odours and visual stimuli: common odours are drawn from a battery of 80 (coffee, spearmint, peanut butter, etc.)
McKeown et al., 1996 [14]										UPSIT Identification + Discrimination: 16-item, three-choice test of odour
Chen et al., 2006 [12]		3.35 ± 2.231		5.63 ± 1.716		7.23 ± 1.236		7.33 ± 1.075		The San Diego Odor Identification Test Threshold Identification
Nijjar and Murphy, 2002 [18]										Threshold: odorant n-butyl alcohol + Task orientation: to determine whether the subject had a good comprehension of the experimental task + Threshold Administration: A two-alternative (stimulus, blank), forced-choice, ascending method, a modified version (Murphy et al., 1990 [15]) as described in (Cain et al., 1983 [51])
Hemdal et al., 1993 [13]										M-UPSIT Identification + Yes/No Identification task + Tactile Identification Task
Zucco and Negrin, 1994 [20]										Matching task: smelled for about 4 s and odour randomly chosen from the set of 10 + Naming task: sniff an odour randomly chosen among the set of 10 for about 4 s while the experimenter read aloud four alternative verbal labels

T-21 = individual with trisomy 21, HC = healthy control, M = molar, UPSIT = University of Pennsylvania smell identification test, M-UPSIT = Modified Pennsylvania Smell Identification Test.

**Table 3 brainsci-11-00952-t003:** Summary of the olfactory function in an individual with trisomy 21.

Author	Main Findings	Notes
Cecchini et al., 2016 [11]	Olfactory function (odour threshold, odour discrimination, odour identification) is severely impaired in both young adults (<30 years and older adults (>30 years) with T-21 Ageing is associated with limited loss of olfactory function, especially odour identification Age effect on olfactory deficit is fully expressed at age > 30 years Females with T-21 have less impaired olfactory function than males over a large age range	
Murphy and Jinich, 1996 [16]	Olfactory function (odour threshold, odour identification, and odour recognition) is severely impaired in participants with T-21 Taste threshold task was comparable with HCs	The concentrations detected by the HCs suggested that they were 50 times more sensitive to odour than the participants with T-21
McKeown et al., 1996 [14]	Olfactory function was comparable among the three groups (T-21, HC, and participants with similar mental capacity with T-21) (Mean age of the participants: 13.89 ± 1.98 years)	
Chen et al., 2006 [12]	Olfactory function (odour threshold and odour identification) is impaired in participants with T-21	Nasal health is comparable in T-21 and HC, and nasal dysfunction is unlikely to contribute to olfactory impairment in participants with T-21 Although participants with T-21 trended toward upper-respiratory infections, sleep-disordered breathing, and nasal itching, differences were not significant
Nijjar and Murphy, 2002 [18]	Olfactory function (odour threshold) is impaired in both young adults (23.2 ± 3.95) and older adults (38.8 ± 7.07) with T-21 Olfactory function (odour identification) is impaired in both T-21 and participants with similar mental capacity with T-21 (Lexical odour identification task is a cognitively demanding, involved memory storage)	Older adults with T-21 performed more poorly than young adults or children Minimal differences in odour thresholds were noted between the three groups at the child age level. These findings, deficits in odour sensitivity are apparent within the participants with T-21 with advancing age
Hemdal et al., 1993 [13]	Olfactory function (odour identification) is impaired in participants with T-21 compared to HCs and participants with similar mental capacity with T-21	Accuracy of identification on the M-UPSIT correlated inversely with age in participants with T-21 only
Zucco and Negrin, 1994 [20]	Olfactory function (odour identification and odour recognition) were deficient in both group: adults with T-21 (41.0 ± 6.8) and younger participants with T-21 (26.5 ± 3.0) Older participants with T-21 score worse in both tasks (matching and naming olfactory task) Younger and older participants with T-21 shows more pronounced impairment in matching task	

T-21 = individual with trisomy 21, HC = healthy control, AD = Alzheimer´s disease.

**Table 4 brainsci-11-00952-t004:** Summary of the finding of olfactory dysfunction in an individual with trisomy 21 and the relation to Alzheimer´s disease.

Author	Conclusion	Notes
Cecchini et al., 2016 [11]	Olfactory function is overall severely impaired in T-21 and maybe globally impaired at a relatively young age, despite the reportedly normal smell	
Murphy and Jinich, 1996 2016 [16]	Olfactory dysfunction may provide a sensitive and early indicator of the deterioration and progression of the brain in older people with T-21	
McKeown et al., 1996 [14]	Olfactory dysfunction in participants with T-21 occurs only at ages when Alzheimer’s-disease-like pathology is present	
Chen et al., 2006 [12]	Olfactory dysfunction in participants with T-21 appears to be secondary to central, rather than rhinology and pathology	
Nijjar and Murphy, 2002 [18]	Olfactory dysfunction in participants with T-21 may be useful in signalling incipient dementia	Older adults with T-21 is a group of persons at risk for AD because of T-21, olfactory impairment is more significant in older individuals, suggesting progressive impairment over time
Hemdal et al., 1993 [13]	Olfactory dysfunction (specific olfactory identification) impairment in participants with T-21 similar to that seen in Alzheimer’s disease	
Zucco and Negrin, 1994 [20]	Olfactory dysfunction was related to the pathological changes in the olfactory epithelium (neuritic plaques and neurofibrillary tangles)	These two olfactory tasks (matching and naming olfactory task) could represent a useful non-invasive diagnostic method

T-21 = individual with trisomy 21, HC = healthy control, AD = Alzheimer´s disease.

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
