# Peer review of "Ageing and Olfactory Dysfunction in Trisomy 21: A Systematic Review"

_brainsci, 2021, doi:10.3390/brainsci11070952_

Round 1
Reviewer 1 Report
This manuscript is a systematic literature review of studies assessing olfactory function in Down syndrome patients throughout life. The authors find that olfactory dysfunction is present in DS patients at as early an age as 30. This is an interesting and clinically important topic that is worthy of review and further research. Two concerns need to be addressed before this manuscript can be published.
- The introduction jumps between a few topics- prevalence of DS, increased prevalence of AD in DS, the concept of advanced aging in DS, then olfactory dysfunction in AD with a comment on olfactory dysfunction in AD models. Since the authors find that dysfunction is present at an earlier age than would be predicted based on AD pathology, which is only beginning to accumulated by the time patients are 30, I think that the impact of this section would be larger if the logic were a bit clearer: 1) DS is a prevalent condition 2) DS patients have impaired olfactory function 3) that dysfunction may be due to trisomy or more specifically to accelerated aging or AD 4) to better understand olfactory dysfunction in DS, we have performed a systematic review of the literature.
- Cecchini et al, 2016, suggests that younger patients are severely impaired. How is this report reconciled with McKeown et al and Nijjar and Murphy that make the case that olfactory function is comparable among T-21 and non T-21 subjects and worsens with age? If there is evidence that impairment worsens with progressive age, then an argument for a role for AD pathology can be made, but if the Cecchini report is accurate, then that might not be so clear. Is the discrepancy due to the way the studies were performed? The numbers of subjects of various ages? The Cecchini article admit to not including cognitive ability of subjects in their study, could that be a cause of increased impairment in young DS patients? This seems to be a critical issue in the interpretation of the findings that should be addressed in this review.
Author Response
Dear Reviewer,
Thank you very much for the suggestions and comments.
Attached are the amendments and corrections we did to improvise the manuscript as per the reviewers' suggestions.
No. |
Comments and Suggestions for Authors
|
Response |
Reviewer 1 |
||
|
This manuscript is a systematic literature review of studies assessing olfactory function in Down syndrome patients throughout life. The authors find that olfactory dysfunction is present in DS patients at as early an age as 30. This is an interesting and clinically important topic that is worthy of review and further research. Two concerns need to be addressed before this manuscript can be published.
1. The introduction jumps between a few topics- prevalence of DS, increased prevalence of AD in DS, the concept of advanced aging in DS, then olfactory dysfunction in AD with a comment on olfactory dysfunction in AD models. Since the authors find that dysfunction is present at an earlier age than would be predicted based on AD pathology, which is only beginning to accumulated by the time patients are 30, I think that the impact of this section would be larger if the logic were a bit clearer: 1) DS is a prevalent condition 2) DS patients have impaired olfactory function 3) that dysfunction may be due to trisomy or more specifically to accelerated aging or AD 4) to better understand olfactory dysfunction in DS, we have performed a systematic review of the literature.
2. Cecchini et al, 2016, suggests that younger patients are severely impaired. How is this report reconciled with McKeown et al and Nijjar and Murphy that make the case that olfactory function is comparable among T-21 and non T-21 subjects and worsens with age? If there is evidence that impairment worsens with progressive age, then an argument for a role for AD pathology can be made, but if the Cecchini report is accurate, then that might not be so clear. Is the discrepancy due to the way the studies were performed? The numbers of subjects of various ages? The Cecchini article admit to not including cognitive ability of subjects in their study, could that be a cause of increased impairment in young DS patients? This seems to be a critical issue in the interpretation of the findings that should be addressed in this review.
+ |
Amendment and corrections to the introduction have been done as per the reviewer suggestion.
Information has been added as per the reviewer suggestion. Information is added in results on page 11 and on a discussion on page 14. |
Thank you.

Reviewer 2 Report
Dear authors,
Your manuscript is interesting but there are serious methodological and structural flaws:
INTRODUCTION
- The objective "to evaluate" is not correct in a systematic review study. The authors have not evaluated anything for themselves.
METHODS
Search Strategy and Study Selection:
- The reference to Moher et al. 2009 is old. There is a 2015 reference. Additionally, there is a 2021 PRISMA statement. The authors should review this.
- The search as of January 1, 2021, is obsolete. Authors should update the search.
Inclusion & exclusion criteria:
- In the inclusion/exclusion criteria there is information that the authors have not specified. The search as of January 1, 2021, is obsolete. Authors should update the search.
Study Quality Assessment:
- How is the level of evidence and grade of recommendation classified?
RESULTS
- The authors said they manually searched the references. This search was supposedly carried out on google scholar. Why is this not shown in figure 1? If the authors have done a reverse search this should appear.
- The authors should include the research design in Table 1.
DISCUSSION
The authors have not made a "discussion".
REFERENCES
This section is not subdivided into subsections. This is wrong.
Many bibliographies are obsolete and some citations are incomplete. The bibliographic citations used are more than 5 years old (81,7% not including "results"). Authors should update the "introduction" and "discussion" references.
Some references are incomplete or have errors. The authors should review this section.
The authors have mixed APA and Vancouver citation regulations. You must write the references correctly.
Author Response
Dear Reviewer,
Thank you very much for the suggestions and comments. We appreciate it so much.
Attached are the amendments and corrections we did to improvise the manuscript as per the reviewers' suggestions.
No. |
Comments and Suggestions for Authors
|
Response |
Reviewer 2 |
||
|
INTRODUCTION
METHODS Search Strategy and Study Selection:
Inclusion & exclusion criteria:
Study Quality Assessment:
RESULTS
DISCUSSION The authors have not made a "discussion".
REFERENCES This section is not subdivided into subsections. This is wrong. Many bibliographies are obsolete and some citations are incomplete. The bibliographic citations used are more than 5 years old (81,7% not including "results"). Authors should update the "introduction" and "discussion" references. Some references are incomplete or have errors. The authors should review this section. The authors have mixed APA and Vancouver citation regulations. You must write the references correctly.
|
The word `to evaluate has been changed to summarize.
Reference has been updated accordingly.
We re-do the articles search in both PubMed and Scopus using the exact keywords. We also manually re-check the references using Google Scholar.
Inclusion and exclusion criteria have been updated accordingly.
The search has been updated.
Quality assessment for each paper selected has been added. Attached in Appendix 1
Amendment and details have been added to Figure 1.
The research design has been added in Table 1.
Amendment and improvision of the discussion have been done accordingly.
The references have been revised.
A new citation has been added to the manuscript.
The correction has been done accordingly.
The correction has been done accordingly. Now, the references using APA 7th edition.
|
Thank you.

Round 2
Reviewer 2 Report
Dear authors,
Your manuscript is interesting but there are serious methodological and structural flaws:
INTRODUCTION
The objective "to evaluate" is not correct in a systematic review study.
METHODS
Inclusion & exclusion criteria:
In the inclusion / exclusion criteria there is information that the authors have not specified. For example, there is no chronological criterion, there is no criterion on which designs were excluded, etc. The study cannot be replicated without explaining the criteria well.
REFERENCES
Many bibliographies are obsolete and some citations are incomplete. The bibliographic citations used are more than 5 years old (618% not including "results"). Authors should update the "introduction" and "discussion" references.
Some references are incomplete or have errors. The authors should review this section.